# From A to m^6^A: The Emerging Viral Epitranscriptome

**DOI:** 10.3390/v13061049

**Published:** 2021-06-01

**Authors:** Belinda Baquero-Perez, Daryl Geers, Juana Díez

**Affiliations:** Virology Unit, Department of Experimental and Health Sciences, Universitat Pompeu Fabra, 08003 Barcelona, Spain; d.geers@erasmusmc.nl

**Keywords:** m^6^A, viral infection, epitranscriptomics, RNA modification

## Abstract

There are over 100 different chemical RNA modifications, collectively known as the epitranscriptome. *N*^6^-methyladenosine (m^6^A) is the most commonly found internal RNA modification in cellular mRNAs where it plays important roles in the regulation of the mRNA structure, stability, translation and nuclear export. This modification is also found in viral RNA genomes and in viral mRNAs derived from both RNA and DNA viruses. A growing body of evidence indicates that m^6^A modifications play important roles in regulating viral replication by interacting with the cellular m^6^A machinery. In this review, we will exhaustively detail the current knowledge on m^6^A modification, with an emphasis on its function in virus biology.

## 1. Introduction

Because of their limited coding capacity, viruses completely depend on the host machinery to multiply. This intimate dependency occurs at multiple layers whose components and features we are still far from understanding. A novel layer recently uncovered is the interaction of viruses with the host RNA modification machinery. Since the discovery of the first chemical RNA modification, pseudouridine (Ψ), in 1957 [1], more than 100 different RNA modifications have been described, collectively known as the epitranscriptome. These modifications occur post-transcriptionally mainly in non-coding RNAs (ncRNA), such as transfer RNA (tRNA) and ribosomal RNA (rRNA). Nevertheless, ten different modifications also occur in mRNAs [2], of which some of the most studied include *N*^6^-methyladenosine (m^6^A), *N*^5^-methylcytosine (m^5^C), and 2′-O-methylation (or Nm, where N stands for any nucleotide).

The m^6^A modification, discovered in 1974 [3,4], entails of the addition of a methyl group to an adenosine base at the nitrogen-6 position. The m^5^C modification, discovered in 1958 [5], entails the addition of a methyl group to a cytosine base at the carbon-5 position. Finally, the Nm modification, discovered in 1967 [6], involves the addition of a methyl group to the 2′-hydroxyl group of the ribose moiety.

The m^6^A modification, is a dynamic process that can be reversed in specific contexts [7,8]. For other RNA modifications, the proteins that catalyse the addition of the chemical modification are well established [9], yet, the precise molecular mechanisms and which proteins interact with all these diverse modifications remain unknown for the large majority. A major limitation has been the lack of reliable and technically accessible methods that allow for precise and quantitative detection of RNA modifications. However, recent methodological advances are rapidly changing this situation. Initial analyses of RNA modifications involved the development of enzymatic digestions of RNA coupled to liquid chromatography-tandem mass spectrometry (LC-MS/MS). While this approach allows for accurate quantification of RNA modifications [10], it does not provide any information on their location. Moreover, the purity of the RNA sample is essential for correct quantifications. These two limitations were overcome for some RNA modifications with the rise of second-generation sequencing that allow mapping them at single-nucleotide resolution in a transcriptome-wide manner [11]. In addition, the recent development of third-generation nanopore RNA sequencing now allows for real-time detection of some RNA modifications at single-nucleotide resolution in native full-length RNA sequences [12,13].

All these recent methodological advances have accelerated our knowledge on the functions of RNA modifications in the cell. These include modulation of RNA structure, facilitating nuclear export, translational activation and regulation of RNA stability [14]. In agreement with these key molecular roles, a myriad of fundamental physiological processes is fine-tuned by RNA modifications including cell differentiation [15], brain development [16], embryogenesis [17], fertility [8], the immune system [18] and the circadian clock [19]. As a consequence, it is not a surprise that alterations in the cellular epitranscriptomic landscape are linked to a wide-range of diseases such as cancer, auto-immunity, neurological and genetic disorders [20,21,22,23]. Excitingly, RNA modifications also play key roles in viral infections. Viral RNA genomes and viral mRNAs are chemically modified during infection and these modifications seem to play fundamental roles in viral life cycles [24,25,26,27]. Moreover, viral infections alter the host epitranscriptomic landscape and these global changes have been shown to affect viral replication [28]. Because of methodological limitations, these viral studies have been mainly focused on m^6^A modifications. In this review, we will summarise the current knowledge of the interplay between m^6^A modifications and viral infection. We will also highlight current limitations in the field and propose key future directions.

## 2. *N*^6^-Methyladenosine (m^6^A)

The m^6^A modification is the most abundant internal modification in cellular mRNA transcripts. Around 25% of all cellular mRNA transcripts are thought to be m^6^A-modified, mainly around the translation stop codon and at the 3′ untranslated region (3′UTR) [29,30]. Because of this abundance and of the recent ability to accurately map its location in a transcriptome-wide manner, m^6^A is the most studied RNA modification. In addition to mRNAs, m^6^A modifications are found in long non-coding RNAs (lncRNAs) [31], circular RNAs [32], the 18S rRNA and 28S rRNA [9], primary microRNAs (pri-miRNAs) [33] and small nuclear RNAs (snRNAs) [34].

The regulation of the m^6^A modification landscape in host mRNAs is a dynamic process that involves m^6^A-methyltransferases to add the modification (writers), m^6^A-binding proteins to recognise and bind to m^6^A modifications (readers) and m^6^A-demethylases to remove the methyl mark (erasers) [2] (Figure 1). The addition of m^6^A methylations occurs co-transcriptionally in the nucleus and is primarily catalyzed by the METTL3–METTL14 methyltransferase complex [35]. In this complex, METTL3 serves as the catalytic subunit and METTL14 as the allosteric adaptor that maintains the complex integrity and binds the complex to the target RNA [36]. Consistently, METTL3 knockout in mouse embryonic stem cells led to a near-complete depletion of m^6^A on mRNAs [37]. The METTL3–METTL14 complex also includes different adaptor proteins, of which the most important is Wilms tumor associated protein (WTAP). This protein is required for the localization of the METTL3–METTL14 complex into nuclear speckles, where pre-mRNA splicing occurs, and for the catalytic activity of the METTL3–METTL14 complex [38]. In the absence of WTAP, RNA-binding capacity of the METTL3–METTL14 complex is strongly reduced [38]. The METTL3–METTL14 complex adds the m^6^A modification in mRNAs at the consensus sequence DRACH (D = A, G or U; R = G or A; H = A, C or U) [2]. However, not all DRACH motifs are m^6^A-modified and around 20% of m^6^A modifications occur outside the DRACH motif [39]. To allow reversible regulation, m^6^A modifications can be erased by the m^6^A demethylases fat mass and obesity-associated protein (FTO) [7] and AlkB homolog 5 (ALKBH5) [8]. However, the widespread reversibility of the process has been challenged [40] and the main substrate of FTO has been proposed to be *N*^6^,2′-*O*-dimethyladenosine (m^6^A_m_) [41].

The m^6^A marks in mRNA molecules are primarily recognised by the five YT521-B homology (YTH) domain-containing proteins YTHDF1, YTHDF2, YTHDF3, YTHDC1 and YTHDC2. Structural crystallographic studies revealed that m^6^A is directly recognised and bound by a conserved aromatic cage structure present in the YTH domain of YTH readers [42]. This domain recognises RNA in an m^6^A-dependent manner without sequence selectivity and regardless of the RNA length [43,44]. A recent RNA affinity assay coupled to mass spectrometry analysis also identified YTHDF1–3 and YTHDC1 as m^1^A readers, however, their affinity was lower for m^1^A RNA baits than for m^6^A baits [45]. Although initially it was proposed that each YTH reader plays different roles, recent evidence suggests a model in which YTH readers behave redundantly [46,47,48]. A major effort from multiple laboratories has been done to identify the role of YTH readers. The main findings are, first, YTHDF1 promotes CAP-dependent translation by interacting with the 5′ UTR-associated eIF3 protein [49]. Interestingly, m^6^A modification in 5′UTRs can also promote CAP-independent translation by directly recruiting eIF3 [50]. Second, YTHDF2 is the main YTH reader that regulates mRNA stability. It recruits the CCR4–NOT deadenylase complex, ultimately promoting degradation of m^6^A-modified RNAs [46,51]. Third, YTHDF3 supports the function of both YTHDF1 and YTHDF2 by binding them in an RNA-independent manner [47,52]. It has been suggested that YTHDF3 would be the first reader to interact with m^6^A and that this binding would facilitate the access to YTHDF1 or YTHDF2 [47]. Finally, YTHDC1 affects splicing of m^6^A-containing pre-mRNAs [53] and promotes nuclear export of m^6^A-decorated mRNAs [54] and the helicase YTHDC2, which is primarily but not exclusively expressed in testes, positively regulating translation of structured mRNAs containing m^6^A modifications located in their coding sequence [55]. Moreover, YTHDC2 plays important roles in spermatogenesis [56,57].

Additional m^6^A readers have been recently identified. These include IGF2BP proteins (IGF2BP1, IGF2BP2 and IGF2BP3) [58], eIF3 [50], FMRP, FXR1, FXR2 [59,60,61], SND1 [59] and three heterogeneous nuclear ribonucleoproteins, hnRNPA2B1 [33], hnRNPC [62] and hnRNPG [63]. However, their mode of binding to m^6^A-modified mRNAs and their role remains to be completely understood. Interestingly, the top two motifs bound by FMRP identified by PAR-CLIP-m^6^A-seq [64] are almost identical to the two top motifs bound by SND1 in m^6^A-modified exons [59], the shared motifs being UGGAC and CU(A/U)CG. Intriguingly, FXR1, FXR2, FMRP and SND1 belong to the Tudor domain ‘Royal family’ and contain an aromatic cage which is structurally similar to the one found in YTH readers [65]. Therefore, these proteins might potentially bind m^6^A directly using this aromatic cage. This does not seem to be the case for the three heterogeneous nuclear ribonucleoproteins. No obvious aromatic cage was identified in the tandem RNA recognition motif (RRM) domains of hnRNPA2B1 [66]. Moreover, hnRNPA2B1 displayed slightly higher affinity to unmethylated RNA than m^6^A-modified RNA in isothermal titration calorimetry experiments [66]. This indicates that hnRNPA2B1 is an indirect m^6^A reader, which may require further protein interactors to bind m^6^A sites. Finally, the hnRNPC and hnRNPG proteins also bind indirectly to m^6^A modifications [62,63]. Importantly, their binding occurs in an RNA secondary structure-dependent manner and it has been suggested that hnRNPA2B1 might use this mode of binding as well [66]. The limited literature available on all these unconventional readers suggests that they bind fewer m^6^A sites compared with the promiscuous YTH readers [59,66]. Future crystallographic or nuclear magnetic resonance (NMR) studies will be necessary to further clarify their mode of binding to m^6^A-decorated transcripts.

The location of the cellular m^6^A machinery has also been thoroughly investigated. Although both writers and erasers are mainly nuclear, they are also found in the cytoplasm [67,68,69,70]. A fraction of FTO has been described to shuttle between the nucleus and cytoplasm [71]; moreover, its subcellular localization was cell type-dependent [69]. Regarding the readers, YTHDC1 is exclusively nuclear while YTHDC2 locates both in the nucleus and in the cytoplasm [55]. The rest of YTHDF readers are widely believed to be exclusively cytoplasmic, however, recent studies also detected YTHDF1, YTHDF2 and YTHDF3 in the nucleus of different cell lines [55,70]. The current availability of high quality endogenous antibodies does now allow, at least for some proteins, detailed characterization of their cellular location under physiological and perturbed conditions.

The m^6^A modification plays key roles in mRNA fates. It is described to modulate mRNA structure [62], splicing [53], nuclear export [72], translation [49,73], stability [47,51] and mRNA localization in stress granules [74]. Under physiological conditions, m^6^A affects a variety of processes such as fertility [8], stem cell renewal capability [75], embryogenesis [17], neuronal development [16], adipogenesis [76] and immune regulation [18]. Importantly, dysregulation of the m^6^A RNA methylation pathway has been connected to the development of a range of human diseases [23] including cancer [77], autoimmune diseases [78] and neurological disorders [79]. Moreover, m^6^A modifications have been extensively reported in viral RNAs and shown to regulate viral life cycles [27,80].

Currently, most methods to detect m^6^A modifications transcriptome-wide combine immunoprecipitations with an anti-m^6^A specific antibody coupled to high-throughput sequencing. The most technically accessible technique, *N*^6^-methyladenosine-sequencing (m^6^A-seq) (Figure 2A), developed in 2012 [29,30], consists of an initial chemical fragmentation step in which mRNAs are reduced to 100–200 nucleotides in length. An input sample is then saved as control and the rest of the fragmented mRNAs are subjected to m^6^A-immunoprecipitation. Following reverse transcription and high-throughput sequencing of input and m^6^A-immunoprecipitated samples, bioinformatic analyses similar to those in ChIP-seq are applied to identify regions enriched for m^6^A decorations, which are commonly referred as m^6^A peaks. Major drawbacks of this method include the lack of single-nucleotide resolution, as it solely provides an insight into the location of m^6^A modifications within a window of 100–200 nucleotides, lack of isoform-specificity and m^6^A stoichiometry (m^6^A abundance at a given site), and a low reproducibility. Even when m^6^A-seq was performed in the same cell lines, the m^6^A peaks in mRNAs overlapped only 30% to 60% between different studies [81]. In recent years, several novel techniques with improved m^6^A resolution are driving our knowledge in the mechanisms and functions underlying m^6^A modification. These include antibody-dependent and independent methods. Two antibody-dependent methods use UV-crosslinking of anti-m^6^A specific antibodies to the m^6^A-modified RNA, the photo-crosslinking-assisted m^6^A-sequencing (PA-m^6^A-seq) [82] (Figure 2B) and the m^6^A individual-nucleotide-resolution crosslinking and immunoprecipitation (miCLIP) [39]. PA-m^6^A-seq pinpoints m^6^A within a window of ≈ 23 nucleotides while miCLIP provides single-nucleotide resolution, however, in contrast to m^6^A-seq, miCLIP requires the use of radioactivity during cDNA library preparation.

In the last five years, multiple new methods have been developed to improve the detection of m^6^A modifications. The m^6^A-level and isoform-characterization sequencing (m^6^A-LAIC-seq) [83] allows, after m^6^A-immunoprecipitation of full-length RNAs, quantifying m^6^A stoichiometry in a transcriptome-wide manner. Another recent antibody-independent method, named meCLICK-seq, utilises click chemistry to attach small molecules to any type of m^6^A-methylated RNA followed by specific cleavage and degradation of these RNAs [84]. By comparing cells treated with and without a click-degrader that catalyses the cleavage of RNA, depletion of modified RNA species is identified across the transcriptome. However, none of these two methods offer an insight into the location of m^6^A sites in the transcripts. To overcome this, four new antibody-independent methods named DART-seq [85], MAZTER-seq [86], SEAL-seq [87] and m^6^A-label-seq [88] allow transcriptome-wide m^6^A mapping. DART-seq (Deamination Adjacent to RNA modification Targets) enables global m^6^A profiling using as little as 10 ng of total RNA to estimate m^6^A abundance in individual RNAs. This elegant technique is based on the fusion of the cytidine deaminase APOBEC1 to the m^6^A-binding YTH domain of YTHDF2. Transfection of the APOBEC1–YTH fusion protein in cells induces C-to-U deamination at sites adjacent to m^6^A modifications and after isolation of total RNA these editing events are identified via standard RNA-seq. MAZTER-seq is a new enzymatic method that takes advantage of the bacterial RNase MazF, which cleaves the RNA immediately upstream of an unmethylated ACA sequence but not an m^6^A-CA sequence. MAZTER-seq offers quantification of m^6^A stoichiometry at single-nucleotide resolution but only at 16–25% of all m^6^A methylation sites. SEAL-seq is an FTO-assisted chemical labeling method that shows high false-negative and high false-positive rates compared with other m^6^A sequencing methods. In this technique, first, the RNA demethylase FTO is expressed in bacteria and purified. Then, the RNA of interest is fragmented and treated with FTO, which oxidises m^6^A to the unstable *N*^6^-hydroxymethyladenosine (hm^6^A). Second, dithiothreitol (DTT) is added to convert hm^6^A to the more stable *N*^6^-dithiolsitolmethyladenosine (dm^6^A). The free sulfhydryl group present in dm^6^A then allows for biotin labelling and purification of the dm^6^A-containing RNAs with streptavidin beads. Recovered RNAs are finally subjected to library construction and deep-sequencing. Finally, m^6^A-label-seq is a metabolic labeling method that detects m^6^A at base resolution and is superior in characterising clustered m^6^A sites. In this method, cells are fed with a methionine analog which substitutes the methyl group on the METTL3 cofactor S-adenosyl methionine (SAM) with the allyl group. Cellular RNAs are therefore metabolically modified with *N6*-allyladenosine (a6A) instead of m^6^A modification. Following iodination-induced cyclization, a6A is converted to Cyc-A which during reverse transcription induces misincorporation at the opposite site in cDNA, thus following deep-sequencing A-to-C/T/G mutations can be identified. Another recent exciting approach to obtain antibody-independent localization of m^6^A modifications is based on nanopore sequencing [89,90]. In this technology, protein nanopores are embedded into a synthetic membrane. When an ionic current is applied and RNA molecules move through these nanopores, the corresponding disruption of the current intensity provides information to identify both the sequence and the modified RNA nucleotide. This approach offers single-nucleotide specificity, isoform-specificity and very long reads without the need of a reverse transcription step. To date, three studies have already used this technology to decipher the epitranscriptome of viral RNA genomes [91,92,93].

## 3. m^6^A Modifications and Viral Replication

Early studies carried out more than 40 years ago identified m^6^A modifications in simian virus 40 (SV40) [94], adenovirus [95], influenza A virus (IAV) [96,97], herpes simplex virus [98], avian sarcoma virus [99] and Rous sarcoma virus (RSV) [100,101]. Since then, m^6^A modifications have been found in many other viral mRNAs, transcribed from DNA and RNA viruses, and retroviruses and in viral RNA genomes. To date, members of the *Flaviviridae*, *Orthomyxoviridae*, *Paramyxoviridae*, *Retroviridae*, *Togaviridae*, *Picornaviridae*, *Polyomaviridae*, *Hepadnaviridae*, *Adenoviridae*, *Rhabdoviridae*, *Herpesviridae* and *Coronaviridae* viral families (Table 1) have been reported to contain m^6^A modifications. Most intracellular steps of the viral life cycle have already been described to be affected by m^6^A modifications (Figure 3). These effects were pro- or anti- viral depending on the virus (Table 2). To address the role of m^6^A modification in viral RNAs, the general approach has been to manipulate the key components of the m^6^A cellular machinery via overexpression or depletion, or to mutate the DRACH motifs within the viral RNAs to abolish specific m^6^A modifications. It is important to note that in the overexpression and depletion studies, the observed effects on virus replication might be indirect as the cellular RNA metabolism will inevitably be affected. In turn, altering the DRACH motif at a specific location in viral RNAs might cause effects beyond those caused by abrogating m^6^A RNA modification as genetic information in viral genomes is highly compacted. Thus, to ascertain a role of specific m^6^A modifications in the viral RNAs, at least a combination of these two approaches would be needed. However, to date few studies incorporate mutational testing within the DRACH motifs in viral mRNAs or viral RNA genomes. Of note, the effect of altering the m^6^A machinery on viral infection might change depending on the cell type used, indicating the high complexity of m^6^A epitranscriptomic regulation during viral infections [102]. The next sections will summarise in detail the current findings on the role of m^6^A modifications in both RNA and DNA viruses.

## 4. Epitranscriptomic Regulation of RNA Viruses

Most studies on the role of m^6^A modifications in RNA viruses focused on single-stranded RNA viruses. These include the positive-sense RNA dengue (DENV), zika (ZIKV), West-Nile (WNV) and the hepatitis C (HCV) viruses, all in the *Flaviviridae* family, the negative-sense RNA influenza virus (IAV) and respiratory syncytial (RSV) viruses and the retrovirus human immunodeficiency virus serotype 1 (HIV-1). All these viruses except IAV and HIV-1, carry out their viral cycles exclusively in the cytoplasm. This poses an intriguing question as both the writing and erasing m^6^A RNA modification machineries are believed to function mainly in the nucleus and re-localization of the involved enzymes into the cytoplasm was not observed in flavivirus or RSV infections [67,68,70]. However, recent studies in enterovirus 71 (EV71) and in human metapneumovirus (HMPV)-infected cells demonstrated that writers and erasers relocate from the nucleus to the cytoplasm [112,113]. Moreover, in HMPV infection, METTL14 strongly co-localised with the viral N protein in cytoplasmic inclusion bodies [113], the sites of HMPV replication. Further confocal microscopy analyses are thus required to determine whether the key RNA-modifying enzymes concentrate in other RNA viral replication factories in the cytoplasm.

In line with a complex RNA virus-m^6^A machinery interplay, currently there is no global pro- or anti-viral role of m^6^A modification that can be generalised. Current data support a pro-viral role for some viruses such as HIV-1 and IAV, and an anti-viral role for others, such as HCV, ZIKV and porcine epidemic diarrhea virus (PEDV) (Table 2).

### 4.1. m^6^A and HIV-1

The HIV-1 genome consists of one single-stranded positive-sense RNA that is found as a dimer and is retrotranscribed in the cytosol of the infected cell into one molecule of DNA that ultimately is integrated into the genomic DNA. Once integrated, the viral DNA exploits the nuclear machinery to transcribe the viral mRNAs and the genomic RNA. These will be exported to the cytosol where they will be translated and, in the case of the genomic RNA, encapsidated to produce the new viral progeny.

HIV-1 RNA contains multiple m^6^A modifications [103,104,106]. In addition, HIV-1 infection causes a global increase of m^6^A modifications in host mRNAs [103]. Interestingly, this increase does not require HIV-1 replication per se. Binding of the HIV-1 gp120 envelope protein to the CD4 receptor in T lymphocytes is sufficient to induce the global increase in m^6^A modifications in cellular mRNAs [125]. Whether and how this affects HIV-1 replication and/or the antiviral immunity remains to be clarified. 

The function of m^6^A modifications within the HIV-1 RNA also remains unclear. Both the untranslated and coding regions have been described to carry m^6^A modifications [103,104,106]. To address their function, most of the studies focus on analysing the effect on HIV-1 infection of overexpressing or silencing different enzymes of the m^6^A modification pathway, including writers, readers and erasers. While two studies assigned a positive role to the m^6^A modifications [103,104], others assigned a negative one [105,106,107]. Although the reason of these contradictory results remains unclear, they might be explained by differences in methodology, cell types used or in the specific m^6^A modification pathway enzymes studied. Importantly, one of the studies addressed the role of specific m^6^A modifications. They showed that m^6^A modification of a highly conserved adenosine in the stem loop II region of HIV-1 Rev Response Element (RREIIB) promotes nuclear export and viral replication [103]. These results supported, at least for one specific location, a positive role of m^6^A modification in the HIV-1 life cycle. However, a recent study challenged the functional role of this modified adenosine. Using NMR analysis and a fluorescence polarization binding assay, it was shown that this m^6^A site has little effect on the structure, or on the binding affinity of RREIIB for the Rev arginine-rich-motif (ARM) in vitro [126]. Moreover, none of the other studies mapping m^6^A modifications in HIV-1 have observed this m^6^A site. Future sequencing studies at single-base resolution or independent of m^6^A antibodies may help in clarifying this discrepancy. Remarkably, recent work showed that YTHDF3 proteins are incorporated into HIV-1 particles in a nucleocapsid-dependent manner [107]. Once incorporated, YTHDF3 proteins limit infection in the new target cell, specifically, at the reverse transcription step. In turn, HIV-1 proteases within the virion cleave YTHDF3 proteins to ensure optimal infectivity of the mature virion.

In conclusion, current data support both negative and positive roles for m^6^A modifications and the m^6^A modification machinery in the HIV-1 life cycle. Further work will be required to complete our understanding of this novel HIV-1-host interaction.

### 4.2. m^6^A and the Flaviviridae Family

All positive-RNA viruses, including the members of the *Flaviviridae* family, replicate exclusively in the cytosol. Moreover, in contrast to the genomes of the other viral groups, the single-stranded positive-sense RNA genomes are highly structured and have a triple function. They act as mRNAs to translate the viral proteins, as templates for replication and as genomes for encapsidation. The regulation of these three functions occurs via specific RNA structures and binding of proteins that we do not completely understand yet.

Multiple m^6^A modifications have been detected by m^6^A-seq in the genome of DENV, ZIKV, WNV, YFV, and HCV. The highest density of m^6^A modifications is located in the coding regions of the RNA-dependent RNA-polymerase (NS5) and the protease (NS3) [67]. In HCV, 19 m^6^A peaks and 42 YTHDF-binding sites were identified by m^6^A-seq and PAR-CLIP, respectively. However, only two high-confidence YTHDF-binding sites overlapped with the m^6^A peaks identified by all m^6^A-seq replicates, suggesting that YTHDF readers might also interact with HCV RNA at non methylated sites. Silencing of the METTL3–METTL14 writers or YTHDF1-3 readers increased HCV virus production while silencing of the FTO eraser showed the opposite effect [67]. Further work demonstrated that this effect occurs at the step of virion production or release. Interestingly, analysis of the subcellular localization of YTHDF1-3 proteins by confocal microscopy revealed that these proteins colocalize with the HCV core protein around lipid-droplets, the site of HCV particle assembly [67]. This was the first observation of host m^6^A readers being recruited to specific viral compartments. Whether this is a common feature of all flaviviruses remains to be further explored. Finally, experiments abrogating four identified putative m^6^A modifications in the E1 gene of HCV RNA resulted in enhanced binding of the viral RNA to the HCV core protein by 2-fold compared to the wild type viral RNA. In addition, the mutated viral RNA produced 3-fold more infectious virions in the supernatant than the wild type virus. Taken together, the authors propose that m^6^A modifications in E1 are bound by YTHDF proteins to negatively regulate HCV packaging while the HCV core protein binds to unmodified viral RNA within the E1 region and facilitates packaging and virion exit.

As found for HCV, m^6^A modifications negatively regulate ZIKV infection [68]. m^6^A-seq identified twelve m^6^A peaks in the ZIKV RNA genome. Silencing of METTL3, METTL14 or YTHDF proteins increased ZIKV production while silencing ALKBH5 decreased it [68]. Overall, the studies in HCV and ZIKV uncovered an anti-viral role for the m^6^A modification in flavivirus RNA. Further investigations will be required to establish whether YTHDF proteins act to negatively regulate packaging in all members of the *Flaviviridae*, as observed for HCV. This raises the question of why m^6^A modifications are potentially conserved in flaviviruses RNA genomes. Plausible explanations include that m^6^A modification might regulate specific stages of flavivirus replication or the host antiviral response. Importantly, recent studies indicate that m^6^A modifications affect the innate immune sensing [113,127].

Infection by flaviviruses alters the m^6^A methylome of host mRNAs [28,68]. Some of these changes alter the expression of proteins that affect flavivirus infection such as RIOK3 or CIRBP [28]. Moreover, cellular pathways activated during viral infections, such as innate immunity and the ER stress response, also contributed to the changes in m^6^A modifications of host mRNAs [28]. Due to the small number of common cellular m^6^A peaks differentially methylated after infection in multiple members of the *Flaviviridae* family (DENV, ZIKV, WNV, and HCV), it will be of interest to address whether other RNA viruses from different families also share this manipulation of the m^6^A pathway.

### 4.3. m^6^A and Chikungunya Virus

The genome of the single-stranded positive-sense RNA virus Chikungunya (CHIKV) from the *Togaviridae* family also contains m^6^A modifications [111]. The authors developed a novel method named viral cross-linking and solid-phase purification (VIR–CLASP) to capture interactions between the pre-replicated viral genome and cellular proteins. In this technique, 4-thiouridine (4SU)-labeled virus is used to infect unlabelled cells followed by photocrosslinking, cell lysis and purification of the 4SU-labeled viral genomes and interacting crosslinked host proteins with Solid-Phase Reversible Immo-bilization (SPRI) beads. After nuclease digestion of the RNA, the identity of the crosslinked host proteins can be retrieved by mass spectrometry analysis. VIR–CLASP revealed that YTHDF2 and YTHDF3 interact with the CHIKV RNA. Moreover, m^6^A modifications were detected within the first 2000 nucleotides of the 5′-end of the CHIKV genome. Overexpression and depletion studies showed a pro-viral role for YTHDF2 and an anti-viral role for YTHDF1 and YTHDF3, a pattern similarly observed for HIV-1, however the molecular mechanisms behind these phenotypes requires further investigation. It is worth highlighting that CHIKV expresses the viral proteins from genomic and subgenomic viral RNAs. The 11.7 kB-long genomic RNA is partially translated to produce four non-structural proteins. In addition, CHIKV synthesises a 4.1 kB subgenomic RNA, identical to the 3’region of the genomic RNA, that gives rise to the structural proteins. Future studies mapping RNA modifications in the genomic and subgenomic RNAs will be necessary to establish whether they are differently modified to carry out distinct functions during the complex life cycle of these viruses.

### 4.4. m^6^A and Coronaviruses

The genome of coronaviruses consists of a single-stranded positive-sense RNA molecule. As CHIKV, coronaviruses also produce subgenomic RNAs. The first identification of m^6^A modifications in a coronavirus was carried out in the porcine epidemic diarrhea virus (PEDV) [108], shortly followed by SARS-CoV-2 [91]. PEDV does not infect humans but causes high mortality associated with severe diarrhea and vomiting in piglets younger than one week old. m^6^A-seq revealed that the PEDV genome isolated from purified viral particles contains seven m^6^A peaks, most of them located in ORF1b, which encodes nonstructural (NSP) proteins. Depletion of m^6^A writers, YTHDF1 or YTHDF2 increased PEDV replication while FTO depletion decreased it. Initial nanopore sequencing analyses suggested that m^6^A might not be one of the major modifications present in SARS-CoV-2 [91]; however, in a recent study which combined m^6^A-seq and miCLIP analyses, 8 m^6^A modifications were identified at single-base resolution in Vero cells [110]. A drawback of this study is that these experiments were carried out using fragmented total RNA, thus they do not allow to locate m^6^A sites to the subgenomic viral RNAs or the region of the genomic RNA that overlaps with these subgenomic RNAs. Depletion of METTL3 or METTL14 led to increased viral replication, while ALKBH5 knockdown had the opposite effect. Of the three YTHDF readers, only depletion of YTHDF2 affected viral replication, which was increased compared with control cells. Therefore, and as found for HCV, ZIKV and PEDV, m^6^A modifications negatively regulate SARS-CoV-2 infection.

### 4.5. m^6^A and Enterovirus 71

The positive-sense single-stranded enterovirus 71 (EV71) RNA genome contains clear m^6^A peaks detected by m^6^A-seq in the genes of the structural VP3 and VP1 and the helicase 2C [112]. Intriguingly, in contrast to that observed for other viral infections, expression and location of m^6^A writers, erasers and readers were altered upon EV71 infection, highlighting a powerful host epitranscriptomic manipulation by EV71. In Vero cells, depletion of METTL3, YTHDF2 and YTHDF3 decreased EV71 replication while FTO knockdown had the opposite effect. Moreover, single point mutation of m^6^A sites in VP1 or 2C resulted in decreased viral titers, indicating a clear positive role for m^6^A in EV71 infection [112]. However, in RD cells, depletion of YTHDF1-3 readers increased EV71 replication [112]. The reason of these divergent m^6^A-mediated responses to EV71 infection in different cells lines remains unknown. Interestingly, mass spectrometry analysis revealed an interaction between METTL3 and the viral RNA-dependent RNA polymerase (RdRp) 3D. Moreover, METTL3 enhanced sumoylation and ubiquitination of the RdRP 3D to facilitate viral replication. The physical and functional interaction of the writer METTL3 with the viral polymerase opens new perspectives in the interplay between the m^6^A machinery and the viral life cycle.

### 4.6. m^6^A and Influenza A Virus

IAV is known to contain m^6^A methylations since 1976 [96], however the location of these and their implications for viral replication remained elusive for decades. IAV contains a segmented genome of eight single-stranded RNA molecules of negative polarity (vRNAs). The IAV genome replicates in the nucleus and expresses the viral proteins in the cytoplasm via expression of RNAs of positive polarity that function as mRNAs. In 2017, a study using PA-m^6^A-seq mapped m^6^A modifications on both the plus (mRNAs) and minus (vRNAs) strands on multiple locations of HA, NP, M, NA and at lower levels on PB1, PB2 and PA viral segments [109]. METTL3 knockout caused an eight-fold reduction in expression of IAV genes which was accompanied by a significant reduction in the production of IAV virions. Additionally, overexpression of YTHDF2, but not YTHDF1 or YTHDF3, increased IAV gene expression and viral replication. PAR-CLIP sequencing showed a correlation between YTHDF1-3 binding sites and the viral m^6^A peaks in the positive and negative strands. Moreover, mutation of multiple m^6^A motifs in the HA gene reduced HA mRNA and protein expression without affecting the expression of other viral mRNA and proteins. Importantly, these viral mutants show reduced IAV pathogenicity in mice. Although the mutations did not affect the amino acid composition, the effect of mutating so many nucleotides at once in the viral RNA structure was not evaluated. This is especially important as m^6^A is known to regulate secondary RNA structure and RNA-binding protein interactions [62]. An additional remaining question is how YTHDF readers can access the m^6^A sites in the vRNA segments, which are tightly bound by nucleocapsid (NP) protein.

### 4.7. m^6^A and Respiratory Syncytial Virus

A recent study revealed with m^6^A-seq that the single-stranded negative sense RSV genomic RNA (gRNA), complementary RNA (cRNA) and the multiple viral RNA transcripts are internally m^6^A-modified in HeLa and A549 cells, with the majority of the m^6^A peaks conserved between the two cell lines [70]. The G viral transcript encoding the attachment glycoprotein present in the surface of the RSV virion was the most extensively modified of all 10 viral mRNA transcripts. m^6^A writers positively regulated RSV replication while m^6^A erasers showed the opposite effect. Overexpression of YTHDF1-3 proteins significantly increased RSV protein expression, gRNA, mRNA synthesis and virion production. Synonymous mutations of conserved m^6^A motifs under three m^6^A peaks in the G transcript showed reduced replication in A549 cells, primary human airway epithelial cultures and respiratory tracts of cotton rats. Together, these results consistently argue for a positive role of m^6^A in RSV infection. Moreover, RSV infection significantly altered the host epitranscriptom; however, in this case the effect was heavily dependent on the cell type analysed, with 2356 m^6^A peaks differentially methylated in HeLa cells and only 44 peaks in A549 cells. The reason of these differences between cell lines remains unknown. It might be plausible that differences in the host cell shut-off dynamics might affect the m^6^A peak calling.

### 4.8. m^6^A and Vesicular Stomatitis Virus

The vesicular stomatitis virus (VSV) is a negative-sense RNA virus from the *Rhabdoviridae* family. miCLIP experiments revealed that the VSV positive-sense RNAs contain 18 m^6^A sites [114]. During VSV infection, METTL3, but not METTL14, relocated from the nuclei to the cytoplasm where it modified VSV (+) RNAs and decreased viral dsRNA formation. Moreover, METTL3 knockdown increased the dsRNA levels on m^6^A sites of VSV RNA. Consequently, the ability of innate RNA sensors, such as RIG-I and MDA5, to bind viral dsRNA was suppressed and the activation of the innate antiviral response prevented. Interestingly, depletion of METTL3 enhanced type I IFN expression and protected mice against VSV infection. These findings highlight METTL3 as a potential target for antiviral intervention.

## 5. Epitranscriptomic Regulation of DNA Viruses

Most DNA viruses replicate in the nucleus and thus have access to the writing and erasing m^6^A machinery. Consistently, the mRNAs expressed from DNA viruses contain m^6^A modifications. As observed for RNA viruses, no global pro- or anti-viral role for viral m^6^A modifications can be generally ascribed to all DNA viruses studied to date (Table 2).

### 5.1. m^6^A and Simian Virus 40

In 1975 it was already reported that the mRNA from the simian virus 40 (SV40), a small DNA tumor virus from the *Polyomaviridae* family, was m^6^A-modified [94]; however, the functional importance of this observation was not elucidated until recently. Using PA-m^6^A-seq, two m^6^A clusters were mapped in the SV40 early region and 11 m^6^A clusters in the SV40 late region [115]. Additionally, PAR-CLIP revealed that most m^6^A clusters on SV40 transcripts are bound by YTHDF2 and YTHDF3. Overexpression of YTHDF2 increased SV40 replication, while knockout of YTHDF2 or METTL3 decreased SV40 replication, indicating a positive role for the m^6^A modification in SV40 replication. Notably, synonymous mutations of the 20 potential DRACH consensus sequences within the 11 m^6^A clusters present in the SV40 late region suggested a non-canonical mechanism of m^6^A modification. Only three of the 11 m^6^A clusters were entirely lost in the mutant virus (VPm), while six m^6^A clusters showed reduced m^6^A levels and two m^6^A clusters were unaffected. This argues that some m^6^A sites in viral RNA may be added independently of METTL3 and/or that m^6^A modifications may occur outside of the DRACH motif more frequently in viral RNA than in host mRNAs. Importantly, the VPm mutant virus replicated more slowly and produced smaller plaques than wild-type (WT) virus. When the VP1 ORF (which contains most of the SV40 m^6^A clusters) was expressed from a plasmid either in WT version or with the mutated m^6^A sites, a 10-fold lower protein expression was seen in the mutant VP1 compared with the WT, without a change in mRNA abundance or the encoded protein sequence, pointing to a deficient mRNA nuclear transport and/or translation.

### 5.2. m^6^A and Herpesviruses

A general feature of viruses within the *Herpesviridae* family is that they cause both lytic (acute) and latent infections. This adds another layer of complexity in the *Herpesviridae* family as two different types of viral transcripts are produced in latently or lytically infected cells. At large, m^6^A modifications regulate latent and lytic viral transcripts of members of the *Herpesviridae* family, as well as viral oncogenesis. Although initial reports carried out decades ago showed an internal m^6^A modification in herpes simplex virus type 1 (HSV-1) RNA [98], within the *Herpesviridae* family, the role of m^6^A in viral RNAs from herpesvirus-infected cells has only been extensively studied in the human oncogenic viruses Kaposi’s sarcoma-associated herpesvirus (KSHV) and Epstein-Barr virus (EBV).

In the latent stage, KSHV is dormantly present as an episome in the host nucleus and expresses only a few genes to sustain latency. Reactivation of KSHV induces the lytic infection which is triggered by the production of the viral master regulator *ORF50* mRNA transcript that encodes the replication and transcription activator (RTA) protein. To date, three different groups have mapped m^6^A modifications in KSHV transcripts with the use of m^6^A-seq [59,102,117]. A comparative analysis of m^6^A peaks detected in both viral and cellular transcripts between all three studies showed an excellent overlap between m^6^A peaks, especially in KSHV mRNAs when the same cell type was compared [59].

The role of METTL3 and FTO has been evaluated by two independent groups in BCBL-1 cells, a primary effusion lymphoma B-cell line [59,116]. In both studies, FTO depletion increased m^6^A levels in KSHV mRNAs and enhanced lytic mRNA and protein expression, while METTL3 depletion had the opposite effect, indicating a pro-viral role for m^6^A during lytic KSHV infection in this cell line. In addition, YTHDF1, YTHDF2 or YTHDF3 depletion also decreased KSHV lytic replication in the same cell line [59]. In contrast, in TREx BCBL-1 cells, which are BCBL-1 cells that contain a Myc-tagged version of spliced viral RTA under the control of a doxycycline-inducible promoter, METTL3 and YTHDF2 knockdown resulted in an increase in RTA protein expression, however, despite the increase in this key viral lytic protein, virion production was not affected [102].

Regarding the role of YTHDF proteins in KSHV replication, contrasting results are also found even when studying the same cell type. Depletion of YTHDF2 in iSLK.BAC16 cells, a renal carcinoma cell line infected by recombinant KSHV BAC16, led to enhanced KSHV replication and increased half-lives of viral transcripts, suggesting an anti-viral role for YTHDF2, while the other YTHDF readers did not consistently affect KSHV replication [117]. However, depletion of METTL3 and YTHDF2 in iSLK.219 cells (iSLK cells infected with the KSHV strain rKSHV.219) decreased RTA protein levels and significantly reduced virion production [102], while the other YTHDF readers did not affect KSHV replication. The observation of both pro-viral and anti-viral roles for YTHDF2 in almost identical cell lines is thus again an issue to take into account.

Mapping of m^6^A sites in KSHV and host transcripts from different cell lines was carried out in an impressive m^6^A-seq study with five different KSHV latently infected cell lines and two cell lines undergoing lytic replication [117]. Moreover, the changes induced by KSHV in the cellular epitranscriptome were also studied in detail by comparing four pairs of uninfected and latently infected cell lines. Intriguingly, it was revealed that KSHV infection triggered 5′UTR hypomethylation and 3′-UTR hypermethylation of host mRNAs, especially in those transcripts involved in transformation pathways, suggesting that KSHV may alter the host epitranscriptomic landscape to induce cellular transformation. This study opened up the opportunity to identify the role of specific m^6^A modifications in cellular mRNAs in the bi-phasic KSHV life cycle, including assessing their role in oncogenic transformation pathways.

More recently, the use of a novel m^6^A peak-calling algorithm [128] facilitated the mapping of m^6^A modifications in KSHV mRNAs by m^6^A-seq in TREx-BCBL-1 cells [59]. To assess which proteins bind to an m^6^A-modified hairpin identified in open reading frame 50 (*ORF50*) mRNA, RNA affinity coupled to mass spectrometry analysis was performed with a methylated *ORF50* RNA hairpin and an unmethylated control. Interestingly, in addition to YTH readers, the m^6^A-modified hairpin was specifically bound by several Tudor-containing proteins, including staphylococcal nuclease domain-containing protein 1 (SND1), all of which contain an aromatic cage structurally similar to the one found in the YTH domain [65,129]. RIP-seq confirmed a strong binding affinity of SND1 for the *ORF50* mRNA and revealed a specific overlap between the RNA-binding sites of endogenous SND1 with cellular m^6^A peaks, uncovering SND1 as a novel m^6^A reader. Importantly, SND1 binding to *ORF50* mRNA was dependent on the methylated status of this RNA and SND1 depletion led to a marked inhibition of KSHV early expression, including the *ORF50* RNA. Moreover, another recent study also associated SND1 with m^6^A modification and showed that SND1 could alter m^6^A levels in colorectal cancer cell lines [130].

In summary, all KSHV m^6^A epitranscriptomic studies to date have reported m^6^A modifications in *ORF50* RNA, highlighting the importance of this highly regulated RNA for KSHV lytic replication. Future experiments using long-read sequencing should help distinguish where the multiple m^6^A sites are located in both the spliced and unspliced *ORF50* RNA and the exact role of m^6^A in this key transcript during lytic reactivation. Moreover, single-nucleotide mapping of m^6^A sites in both viral and cellular mRNAs during latent and lytic infection coupled to mutagenesis of specific m^6^A sites will allow further gain and loss of function studies to elucidate how m^6^A influences KSHV replication and how KSHV triggers the oncogenic pathway.

Another member of the *Herpesviridae* family was recently reported to harbour m^6^A modifications is Epstein–Barr virus (EBV) [118,119]). Using m^6^A-seq, two different groups mapped m^6^A modifications in the latent and lytic EBV mRNAs [118,119]. METTL14 depletion led to reduced stability of latent transcripts, whereas it increased stability of lytic transcripts [118]. Interestingly, METTL14 transcript and protein levels were dramatically increased in EBV latently infected cells. EBV latent antigen EBNA3C positively regulated METTL14 mRNA expression by activating the METTL14 promoter and also stabilising METTL14 protein. Moreover, METTL14 was upregulated in EBV-positive tumors and co-localised with EBNA3C. Notably, knockdown of METTL14 or EBNA3C in NOD-SCID mice led to a dramatic slowing of tumor growth compared to the control group. Overall, these results point at METTL14 as an essential factor in EBV replication and oncogenesis representing an innovative therapeutic target to treat EBV-positive tumors. Another team observed that out of all the m^6^A-related proteins, YTHDF1 depletion promoted EBV infection the most [119]. Mechanistically, YTHDF1 binds the viral lytic transcripts *BZLF1* and *BRLF1*, and YTHDF1 knockdown increased their half-lives [119].

### 5.3. m^6^A and Adenovirus

The nuclear-replicating adenoviral RNAs were reported to bear m^6^A modifications in the 1970s [95]; however, it was only recently found that by using direct RNA long-read sequencing these modifications were located at the nucleotide- and at the isoform-level [93]. As many adenoviral RNAs have overlapping regions, m^6^A-seq failed to accurately map viral m^6^A modifications. Instead, with the combination of wild type and METTL3 KO A549 cell lines and direct RNA sequencing, a total of 97 m^6^A sites were elucidated in the adenoviral serotype 5 (Ad5) RNAs. While depletion of YTHDF1-3 readers or m^6^A erasers did not affect viral replication, METTL3 or METTL14 knockdown reduced viral replication. Specifically, METTL3 knockdown negatively impacted the splicing and accumulation of late adenoviral RNAs [93].

### 5.4. m^6^A and Hepatitis B Virus

The hepatitis B virus (HBV) genome consists of a partially double-stranded molecule of circular DNA that localises into the nucleus. There, HBV DNA transcribes the viral mRNA and pregenomic RNA (pgRNA), which are exported to the cytosol to express the viral proteins. The pgRNA is also used in the cytosol as a template for reverse transcription to generate new molecules of HBV genomes. Recently, m^6^A modifications were detected in HBV mRNAs and pgRNA from HBV-expressing cells and from liver tissues of chronic HBV patients [120]. Depletion of writers YTHDF2 and YTHDF3 increased the expression of the HBV proteins HBc and HBs, whereas FTO or ALKBH5 depletion had the opposite effect, indicating a negative effect of the m^6^A machinery on HBV expression. Depletion of METTL3 and METTL14 writers or of the reader YTHDF2 significantly extended by ≈10 h the pgRNA half-life, indicating that m^6^A modification significantly lowers the stability of HBV RNA transcripts. Moreover, depletion of writers inhibited reverse transcription of HBV pgRNA. A conserved m^6^A consensus motif bearing an m^6^A modification was identified by m^6^A-seq in the epsilon stem loop structure at the 3′-end of all HBV transcripts and at both the 5′-end and 3′-end epsilon stem loops of pgRNA. Using different single point A to C mutant pgRNAs, either lacking m^6^A modifications at the 5′-end, 3′-end or both, revealed that the effect of m^6^A modification depends on the location of m^6^A in the pgRNA. Indeed, m^6^A methylation at the 5′ end of the pgRNA facilitated reverse transcription of pgRNA, while the m^6^A at the 3′ end made HBV RNAs less stable. How m^6^A modification modulates reverse transcription of HBV pgRNA is currently unknown but it is possible that YTHDF readers are involved in this modulation as seen for HIV-1.

## 6. Epitranscriptomic Regulation of the Immune Response to Viral Infection

The m^6^A RNA modification affects the innate and adaptive immune responses. Innate immunity to viral infections heavily relies on the type I IFN response in which foreign viral RNA is recognised by the pathogen recognition receptors (PRR). These include the toll-like receptors (TLR) 3, 7 and 9 in endosomes and the RIG-I like receptors (RLR) in the cytosol of which the most important include the retinoic acid-inducible gene-I (RIG-I) and the melanoma differentiation-associated protein 5 (MDA5). Triggering of endosomal or cytosolic RNA sensors induces a signaling cascade leading to the expression and secretion of IFNα and IFNβ that are recognised by the interferon receptor (IFNAR). This results in the activation of the JAK–STAT pathway and the subsequent transcription of hundreds of interferon stimulated genes (*ISGs*) that mediate an antiviral response [131].

Viruses have evolved to escape or inhibit the type I IFN response by multiple mechanisms. These include the m^6^A methylation of viral RNAs to prevent recognition by the innate RNA sensor RIG-I [113,127]. For example, m^6^A-deficient HMPV RNA facilitated the RIG-I conformational change necessary to enhance downstream IFN signaling [113]. Infection of cotton rats with mutant HMPV viruses lacking multiple m^6^A sites induced significantly higher type I IFN responses in cotton rats than the parental HMPV viruses and resulted in attenuated infection [113]. Moreover, following depletion of METTL3 or YTHDF2 proteins, the mRNA expression of *IFNβ* and various *ISGs* was enhanced in human and murine cytomegalovirus- (HCMV and MCMV), IAV-, adenovirus- or vesicular stomatitis virus (VSV)-infected cells [121]. Similarly, YTHDF2 and YTHDF3 silencing led to an increase in *IFNβ* mRNA levels in cells transfected with HCV RNA carrying a mutation that abrogates viral replication [127]. Moreover, the IFNβ levels were unaffected when cells were transfected with the same HCV RNA that contained a mutation that abrogated an m^6^A modification near a pathogen-associated molecular pattern (PAMP) present in the HCV RNA. Consistently, METTL14 depletion resulted in significantly elevated *IFNβ* mRNA levels in HCMV-infected cells and uninfected cells exposed to dsDNA, while ALKBH5 depletion reduced *IFNβ* mRNA levels [124]. Relevantly, the *IFNβ* mRNA has three m^6^A sites whose abolition stabilises *IFNβ* mRNA, which in turn sustains prolonged expression of the cytokine IFNβ, resulting in a stronger antiviral response [121]. YTHDF3 has also been reported to negatively regulate anti-viral immunity through specific suppression of *ISGs* expression and without affecting *IFNβ* mRNA levels [123]. Mechanistically, YTHDF3 promotes translation of the transcription repressor forkhead box protein O3 (FOXO3), which is known to negatively regulate *ISG* expression. Consequently, silencing of YTHDF3 in macrophages inhibited viral infection with several DNA and RNA viruses, which included VSV, encephalomyocarditis virus (EMCV) and herpes simplex virus type 1 (HSV-1). Importantly, YTHDF3-/- mice were protected against infection with VSV. Recently, it was also found that in response to VSV infection, host cells impair the enzymatic activity of the m^6^A eraser ALKBH5 by inducing R107 demethylation on the ALKBH5 protein [122]. Due to decreased ALKBH5 activity, m^6^A methylation is increased in the α-ketoglutarate dehydrogenase (*OGDH*) transcript. This leads to reduced *OGDH* mRNA and protein expression. As OGDH is an enzyme that produces the metabolite itaconate, required for viral replication, viral replication is inhibited without the involvement of the innate immune system.

Collectively, these data show an exciting interplay between viruses-m^6^A modifications and immune responses that might allow for novel therapeutic interventions.

## 7. Conclusions and Future Perspectives

Most of our current knowledge of m^6^A modifications in viral RNAs have emerged in the past five years, consequently we are only beginning to understand the exact abundance and role of this modification during viral replication. Currently, several high-throughput deep-sequencing m^6^A mapping techniques at single-nucleotide resolution exist; however, most viral studies to date have relied on m^6^A-seq to locate m^6^A in viral RNA, with the exception of the use of PA-m^6^A-seq for IAV, HIV-1, SV40 and EBV, miCLIP for VSV and SARS-CoV2, and the use of nanopore sequencing for adenovirus. Because the resolution of m^6^A-seq is low (100–200 nucleotides) and m^6^A can appear in clusters, which results in overlapping m^6^A peaks, peak-based prediction algorithms miss more than half of m^6^A sites occurring in these clusters [39]. Clearly, the use of single-nucleotide resolution mapping studies has to be the next step forward in understanding m^6^A modifications in viral RNA. Moreover, whether the identified m^6^A peaks in viral RNAs contain DRACH motifs or whether these modifications are METTL3-dependent, the major catalyser of m^6^A addition in host mRNAs remains to be clarified for the majority of viruses investigated to date. This is of particular importance because (i) when DRACH motifs are abrogated in viral RNAs, only a reduction of m^6^A levels is generally observed [70,115] and (ii) other host m^6^A-methyltransferases such as METTL16, METTL5 and ZCCHC4 have been recently identified. METTL16 targets specific host pre-mRNAs, mRNAs, snRNAs and lncRNAs [34]. METTL5 and ZCCHC4 exclusively methylate a single m^6^A site present in the 18S rRNA and in the 28S rRNA, respectively [132,133]. Therefore, it is feasible that viral RNAs might be methylated by enzymes other than METTL3, especially in viruses with non-canonical mRNAs, such as the very long and highly structured RNAs from flaviviruses and coronaviruses.

Abrogation of m^6^A in viral RNAs led to reduced viral replication in the vast majority of viruses investigated, indicating a general positive role for m^6^A in viral RNA. However, it is important to note that only one study in HBV [120] confirmed that the resulting effect on viral replication was a sole consequence of the loss of m^6^A and not a consequence of altering the viral RNA secondary structure after mutating A to C in the DRACH motif. This is particularly important when several mutations are introduced at once and it should be addressed to gather more solid evidence of the role of m^6^A modification. m^6^A mapping at single-nucleotide resolution will allow pinpointing specific sites to mutate instead of mutating all DRACH sites under a given m^6^A peak. Now that some m^6^A sites have been accurately located in viral RNAs, another step forward would be understanding the putative role of m^6^A in regulating viral RNA secondary structures and elucidating which m^6^A readers bind to these sites. Another critical question is whether m^6^A sites act redundantly in a single transcript, especially in those heavily m^6^A-modified viral RNAs, and whether abrogation of multiple sites is actually required to see a clear effect on viral infection.

The role of the YTHDF readers on viral replication has been extensively evaluated with depletion and overexpression studies in multiple viruses. Both YTHDF pro-viral and anti-viral functions have been reported depending on the virus and cell type used. While initial papers described different functions for each reader, with YTHDF1 enhancing translation of m^6^A-modified RNAs [49], YTHDF2 promoting m^6^A-modified mRNA degradation [51] and YTHDF3 enhancing both translation and mRNA degradation of methylated transcripts [47], these reader functions cannot always be directly translated to viral RNAs. For example, for some viruses YTHDF2 has been shown to mediate viral mRNA degradation [117,120], while in contrast, in other viruses overexpression of YTHDF2 increased viral RNA levels [111]. Moreover, a new model proposes that YTHDF1-3 readers act redundantly to mediate m^6^A-modified mRNA degradation [48]. Although not all viral studies have always evaluated the role of the three different YTHDF readers, it is now clear that YTHDF proteins do not always act redundantly and they clearly exhibit different phenotypes in viral infections. This has been observed in HIV-1, CHIKV, IAV and KSHV, suggesting that although the human C-terminus YTH domain is highly conserved between these proteins [134], the less conserved N-terminus differs and may add an extra layer of regulation that discriminate the role of these proteins on viral RNAs. Moreover, it is important to realise that depletion and overexpression of the m^6^A machinery has a broad effect on cellular homeostasis by affecting host RNA biology. Therefore, any effect on viral replication as a consequence of depleting m^6^A writers, erasers and readers could be an indirect effect from the depletion of crucial players in regulating RNA fates. For example, despite ALKBH5 depletion resulting in marked inhibition of VSV, m^6^A-seq analysis performed on WT and *ALKBH5*-deficient macrophages revealed that there was no up-regulation of m^6^A signal on VSV RNA. Thus, ALKBH5 promotes VSV replication not by targeting m^6^A modification of VSV RNA but by directly targeting m^6^A modification on the host *OGDH* mRNA to up-regulate the OGDH-itaconate metabolic pathway [122]. Finally, besides YTHDF proteins, other m^6^A readers could also play important roles in viral infections as recently shown for KSHV and the SND1 protein [59], hence, further RNA affinity experiments will be of great interest to elucidate the viral m^6^A-interactome.

To date, very few studies have shed light on how virus-induced changes in the host m^6^A landscape impact the viral replication cycle. Intriguingly, despite viral infections causing major perturbations in the cell at multiple levels, the virus-induced remodeling of the host epitranscriptome seems to occur at a limited amount of m^6^A sites [28,59]. Another important aspect to be explored is whether particular m^6^A sites in viral RNA can be subjected to demethylation during a viral infection cycle, and whether FTO or ALKBH5 may play a role in this process. Moreover, most studies have been carried out in transformed cell lines. Therefore, further epitranscriptomic studies using relevant primary cell lines and in vivo studies are essential. Most viruses infect a narrow spectrum of related hosts or cell types. Important exceptions are the emerging mosquito-borne viruses such as DENV, WNV, ZIKV and CHIKV that efficiently infect both humans and mosquitoes; two organisms separated by around one billion years of evolution. Whether the viral RNAs are differentially modified in both hosts and whether the arthropod m^6^A landscape is also altered under viral infection conditions is unexplored. However, under bacterial infection conditions, a recent study in *Drosophila melanogaster* flies showed that infection with the bacteria *Wolbachia pipientis* conveys protection against RNA virus infection by upregulation of the *D.melanogaster* methyltransferase MT2 [135]. MT2 is a m^5^C methyltransferase that catalyses a methylation to the fifth carbon on a cytosine base in tRNA^Asp^. *Wolbachia*-infected flies have a seven to eight-fold increase in *MT2* transcript levels and repress Sindbis virus (SINV) infection. On the other hand, *MT2* knockout flies are more susceptible to SINV infection than wild-type flies. Together, this study suggests an anti-viral function for m^5^C methylation in flies and argues for a possible role for m^6^A methylations in regulating virus infection in other insect vectors. It would thus be of great interest to study the interplay of m^6^A/mosquito-borne viruses in mosquitoes and to compare with that in humans. Putative differences might help understanding why and how these viruses cause an acute infection in human cells and a chronic one in mosquito cells.

The importance of the epitranscriptome in regulating viral replication and antiviral immunity opens the door for novel therapeutic interventions that target either the host or viral epitranscriptome to restrict viral replication. Our current knowledge supports that m^6^A modifications might act in a pro-viral or anti-viral manner depending on the virus, cell type analysed, and even on its location in the viral RNA molecule. Thus, an approach that specifically targets key m^6^A sites in the viral RNAs seems ideal, although quite challenging. A more general approach that enhances m^6^A methylations with specific compounds [136] might be of interest to treat viruses negatively regulated by them. However, these treatments will also affect the host RNAs. Besides drugging the viral epitranscriptome, generation of viral strains with altered m^6^A methylations might be useful for vaccine purposes because m^6^A modifications have been implicated in the antiviral immune response [18] and m^6^A-depleted viruses are known to induce potent type I IFN responses in vivo [113]. Although the functionality of such a vaccine has not been evaluated yet in vivo, it seems plausible that infection with live attenuated vaccines based on m^6^A-depleted viruses might induce potent adaptive immune responses providing the host with long lasting immunity to wild-type viruses. In line with this, RSV viruses expressing G transcripts depleted of m^6^A modification and m^6^A-deficient HMPV viruses were highly attenuated yet retained high immunogenicity in cotton rats [70,113], while IAV viruses depleted of m^6^A modification in HA resulted in reduced IAV pathogenicity in mice [109].

Recent advances point out the host m^6^A machinery as a key interactor in viral infections. Advancing in this novel virus-host interplay is of great interest not only in virology but also in cell biology as viruses have proven to be great tools to uncover novel aspects of multiple cellular processes. In conclusion, the journey of deciphering the viral m^6^A epitranscriptome, which started in the 1970s, has truly bloomed in the past five years and pointed to the host m^6^A machinery as being an important component in viral infections. However, to go beyond descriptive observations and derive a mechanistic understanding of viral RNA modifications and their benefits for virus survival requires further efforts. The recent m^6^A sequencing technologies will clearly be a major help for this endeavor. 

## Figures and Tables

**Figure 1 viruses-13-01049-f001:**
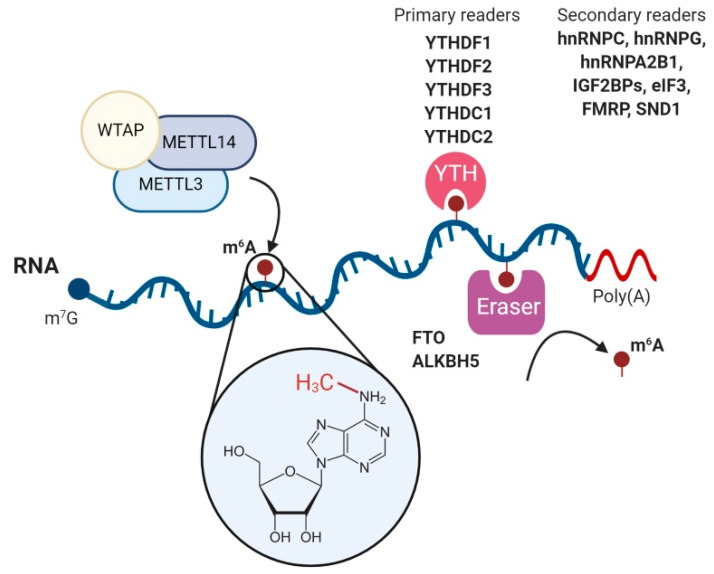
The m^6^A RNA modification and its cellular interactome. The *N*^6^-methyladenosine (m^6^A) modification is installed co-transcriptionally to mRNA transcripts by the METTL3-METTL14-WTAP writer complex and can be removed by the FTO and ALKBH5 erasers. The family of YTH proteins comprises the major readers that specifically recognise and directly bind m^6^A modifications. YTHDC1 is a nuclear m^6^A reader and facilitates mRNA splicing and nuclear export. In the cytoplasm, YTHDF1, YTHDF2, YTHDF3 and YTHDC2 bind mRNAs to enhance their translation or induce their degradation in P-bodies. Other multiple secondary readers that also target m^6^A modifications have been recently elucidated. Some of these bind to m^6^A through structural RNA switches in the mRNA transcript induced by m^6^A itself.

**Figure 2 viruses-13-01049-f002:**
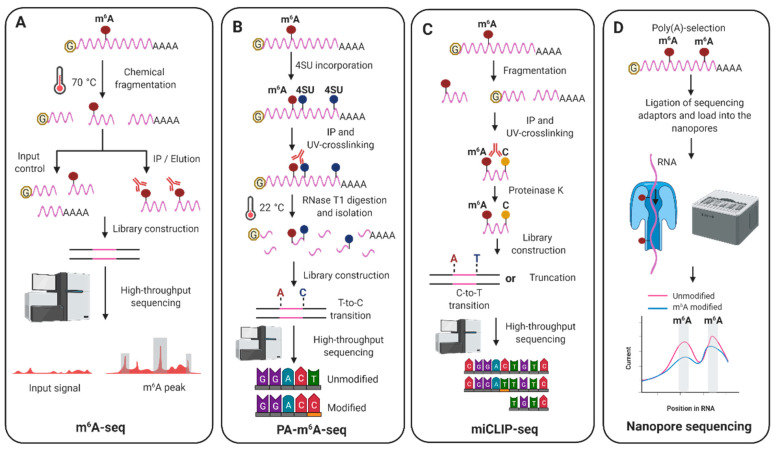
The most common transcriptome-wide m^6^A profiling techniques used to locate m^6^A in viral RNAs. (**A**) The first step of the m^6^A-seq technique consists of the chemical fragmentation of the RNA into 100–200 nucleotides-long RNA molecules. An input control (which does not undergo m^6^A-immunoprecipitation) and an m^6^A-immunoprecipitated (IP) sample are required to build cDNA libraries for deep-sequencing and subsequent bioinformatic analysis. m^6^A-seq offers a low-resolution approach to map m^6^A modifications because of the relatively large size of the starting fragmented RNAs. (**B**) Photo-crosslinking-assisted m^6^A-sequencing (PA-m^6^A-seq) requires incubating cells with 4-thiouridine (4SU), which is incorporated into mRNAs. Full length mRNAs are then immunoprecipitated with m^6^A antibodies and UV-crosslinked. Next, crosslinked mRNAs are digested into around 30 nucleotides-long molecules that are isolated to prepare the cDNA libraries. Crosslinked 4SU is read as C during reverse transcription, therefore T to C transitions can be identified when compared to the reference genome. (**C**) In miCLiP-seq, RNA containing m^6^A-modifications is fragmented, immunoprecipitated (IP) using an anti-m^6^A antibody and UV-crosslinked to the IP antibody. As similarly to the previous techniques, during library preparation adapters are ligated to the precipitated RNA but in this case the 5′-end of the RNA is radioactively labeled. After purification of the RNA-antibody complexes using SDS-PAGE and membrane transfer, the RNA fragments are then reverse-transcribed leading to either a transition of C to T or truncations in cDNA. Following library construction and high-throughput sequencing localization of m^6^A modifications at single nucleotide resolution is achieved. (**D**) Nanopore sequencing of m^6^A modifications takes advantage of changes in electric current when a nucleotide containing a modification traverses a nanopore. Using specialised software, m^6^A modifications can then be identified. This allows for fast, accurate and sensitive detection of long sequencing reads reaching up to 2 Mb.

**Figure 3 viruses-13-01049-f003:**
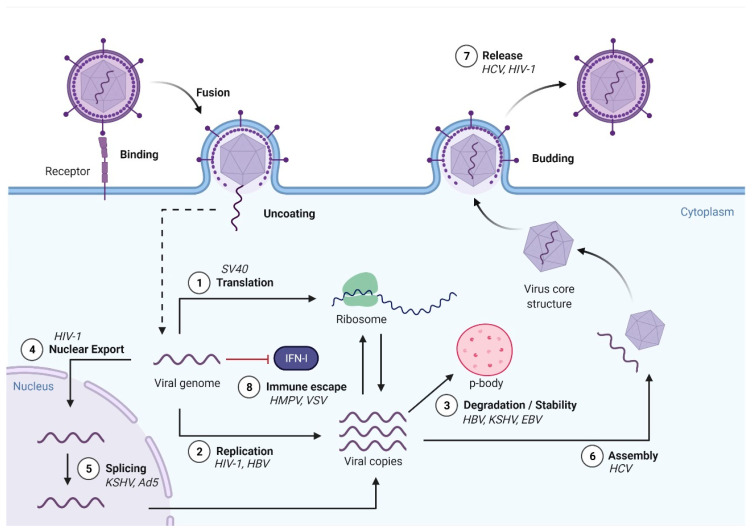
How viral m^6^A modifications and the host m^6^A machinery affect viral replication. Both DNA and RNA viruses are subjected to extensive m^6^A modification. Importantly, m^6^A modifications differentially impact the steps of the viral life cycles depending on the virus. (1) m^6^A modifications enhance the translation of some viral mRNAs. (2) m^6^A modifications prevent replication through the inhibition of reverse transcription in retroviruses and enhance replication of HBV through the facilitation of reverse transcription. (3) m^6^A modifications reduce or increase viral mRNA stability. (4 and 5) m^6^A modifications affect the splicing and nuclear export of some viral RNAs that replicate in the nucleus. (6) The encapsidation of genomic RNA or assembly of viral components can be regulated by m^6^A modifications. (7) m^6^A modifications reduce or increase infectious viral titers. (8) Viral m^6^A decorations can prevent virus-sensing by RIG-I or MDA5 receptors and consequently inhibit type I IFN induction and anti-viral immunity.

**Table 1 viruses-13-01049-t001:** Literature on viral m^6^A epitranscriptomic studies.

Virus	Detection	Phenotype	Main Function for m^6^A	Reference
RNA Viruses				
HIV-1	m^6^A-seq	Proviral	vRNA nuclear export	[103]
	PA-m^6^A-seq	Proviral	vRNA abundance and protein expression	[104]
	m^6^A-seq	Antiviral	YTHDF1–3 proteins inhibit HIV-1 infection by decreasing HIV-1 reverse transcription	[105,106]
	-	Antiviral	YTHDF3 inhibits HIV-1 infection at the step of reverse transcription	[107]
HCV	m^6^A-seq	Antiviral	m^6^A modifications in E1 are bound by YTHDF proteins to negatively regulate HCV packaging	[67]
ZIKV	m^6^A-seq	Antiviral	vRNA abundance and protein expression	[68]
DENV WNV	m^6^A-seq	-	-	[67]
PEDV	m^6^A-seq	Antiviral	m^6^A modifications reduce viral RNA production and viral titers	[108]
IAV	PA-m^6^A-seq	Proviral	m^6^A modifications increase IAV RNA expression	[109]
RSV	m^6^A-seq	Proviral	m^6^A modifications enhance RSV replication and gene expression	[70]
SARS-CoV2	m^6^A-seq and miCLIP	Antiviral	m^6^A inhibits SARS-CoV2 replication	[110]
CHIKV	m^6^A-IP	Proviral and antiviral	YTHDF1 and YTHDF3 restrict CHIKV replication, while YTHDF2 promotes it	[111]
EV71	m^6^A-seq	Proviral	METTL3 induces enhanced sumoylation and ubiquitination of the viral RNA polymerase to facilitate viral replication	[112]
HMPV	m^6^A-seq	Proviral	m^6^A modifications enable vRNA to escape recognition by RIG-I	[113]
VSV	miCLIP	Proviral	m^6^A modifications reduce viral dsRNA formation leading to reduced virus-sensing by innate receptors	[114]
**DNA VIRUSES**				
SV40	PA-m^6^A-seq	Proviral	m^6^A enhances the translation of viral late transcripts	[115]
KSHV	m^6^A-IP	Proviral	Splicing of *ORF50* pre-mRNA	[116]
	m^6^A-seq	Antiviral	vRNA stability of latent and lytic transcripts	[117]
	m^6^A-seq	Antiviral and proviral	vRNA abundance and protein expression	[102]
	m^6^A-seq	Proviral	vRNA abundance and protein expression	[59]
EBV	m^6^A-seq	Both	vRNA stability of latent and lytic transcripts	[118]
	m^6^A-seq and PA-m^6^A-seq	Antiviral	YTHDF1 promotes EBV RNA decay	[119]
Ad5	m^6^A-seq and nanopore	Proviral	m^6^A is required for splicing of adenoviral late transcripts	[93]
HBV	m^6^A-seq	Antiviral and proviral	vRNA stability and viral reverse-transcription	[120]

**Table 2 viruses-13-01049-t002:** Effects observed on viral replication after manipulation of the cellular m^6^A machinery. Boxes shaded in green indicate enhanced viral replication, while in red indicate reduced viral replication. Boxes shaded in orange indicate that there was no significant effect on viral replication. DNA viruses are shaded in grey boxes. HIV-1: [103,104,105,106,107]. HCV: [67]. ZIKV: [68]. CHIKV: [111]. PEDV: [108] IAV: [109,121]. RSV: [70]. SARS-CoV2: [110]. EV71: [112]. HMPV: [113]. VSV: [114,122,123]. SV40: Tsai et al., 2018. KSHV: [59,102,116,117]. EBV: [118,119]. Ad5: [93]. HBV: [120]. HCMV: [121,124].

Virus	Depletion/Knockout													Overexpression									
	METTL3		METTL14		FTO	ALKBH5		YTHDF1		YTHDF2		YTHDF3		METTL3	METTL14	FTO	ALKBH5	YTHDF1		YTHDF2		YTHDF3	
HIV-1	[103,106]		[103,106]		[106]	[103,106]		[105,106]		[105,106]	[104,105]	[105,106,107]						[104]	[105,106]	[104]	[105,106]	[104]	[105,106]
HCV																							
ZIKA																							
CHIKV																							
PEDV																							
IAV	[109,121]																	[109]		[109]		[109]	
RSV																							
SARS-2																							
EV71																							
HMPV																							
VSV	[122]	[114]	[122]		[122]	[122]		[123]		[123]		[123]		[114]			[122]						
SV40																							
KSHV	[102]	[59,102,116]			[59,116]			[102,117]	[59]	[102,117]	[59,102]	[102,117]	[59]							[117]			
EBV	[119]		[118]	[118,119]		[118]	[119]	[119]		[118]	[118,119]	[119]											
Ad5																							
HBV																							
HCMV	[121,124]		[121,124]		[121,124]	[121,124]		[121]		[121]		[121]											

## Data Availability

Not applicable.

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
