# Peer review of "From A to m6A: The Emerging Viral Epitranscriptome"

_viruses, 2021, doi:10.3390/v13061049_

Round 1
Reviewer 1 Report
Admittedly, I was not very enthusiastic at first to peruse yet another review on N6-methyladenosine, as off the top of my hat I could think of two recent review articles on this topic in high-impact journals from the labs of Samie Jaffrey or Chuan He, not to mention the many other more specialized reviews. However, in this article the authors explore this RNA modification from a novel virology-centric angle and thus provide an original and refreshing pitch to this seemingly exhausted treatise. Furthermore, I was skeptical at first about the authors bold claim in the abstract to “EXHAUSTIVELY detail the current knowledge on m6A […] in virus biology”. However, the authors indeed fulfill their bold claim, and in my opinion this review article has the potential to become an essential reference for virologist working on m6A.
Clearly, the strength of this review is its thorough recapitulation of recent literature, incorporating many different viruses and how they take advantage of the m6A system. On the other hand, a minor shortcoming of this review presents itself when the authors describe technical aspects of m6A detection. As this review is targeted towards virologist, it may be beneficial to the reader to include more details about experimental outline. For example, on page 10 it is stated that some “techniques require the use of radioactivity” but it is unclear at which step radioactivity is required. This could be further elaborated on in Figure 2. This information may be particularly helpful for researchers new to the field of epitranscriptomics in selecting a feasible m6A-detection method in their labs. Moreover, for the sake of completeness, the authors should consider including a schematic of miCLIP in Figure 2, as this technique is often reported in the literature, probably more so than PA-m6A-seq. Along these lines, a few additional sentences when describing “novel” antibody-independent methods (How is it novel? page 11, 1st paragraph; page 18 2nd paragraph) would do these clever approaches justice.
Lastly, the manuscript is well written but requires some final polishing, as there are several typos throughout the manuscript, such as on page 2 2nd paragraph, page 19 second to last line, page 20 line 17. One of the typos that should be fixed absolutely before publication is to correctly spell MAZTER-seq (not Master-seq) on page 11.
Author Response
We thank the reviewers for peer-reviewing the manuscript and the kind comments received. We have now addressed their comments in the manuscript and figures to improve the review.
Reviewer 1:
We have now included more experimental details about the latest cutting-edge m6A-mapping techniques.
We have also updated figure 2 to include miCLIP and nanopore sequencing.
We have corrected the typos throughout the manuscript as far as we can see.
We have shown the changes with tracked changes on the document.
Reviewer 2 Report
I have to start my comments by congratulating Baquero-Perez and colleagues for this wonderful state-of-the-art bibliographical revision on the interplay between N6-methyladenosine and viruses. The manuscript is complete and well written providing detailed information on the general aspects of m6A and its associated cellular machinery, the interplay between m6A and viruses, their relationship with the innate immune response and, finally, a very good perspectives section. I really enjoyed the reading and I am sure this manuscript will be of interest for a broad audience, not only virologists.
I only have a some very minor comments to this fantastic manuscript:
1) Last paragraph on page 2 refers to RNA modifications in general thus, to say "is a dynamic process that can be reversed" may be confusing to non-specialized readers since only m6A has been proved to be reversible (still with some concerns on the reversibility of the process). Same paragraph: "Yet, the precise molecular mechanisms and the proteins governing the addition and removal of RNA modifications remain unknown"... it would be better if authors precise that this is true for most RNA modifications but that for some of them there is available information.
2) Page 4: "mainly around the translation stop codon and at the 3′ untranslated region (3′UTR)" should be accompanied by the corresponding references (Dominissini et al, 2012 and Meyer et al, 2012). These articles also reported the presence of m6A in lncRNAs.
3) Last paragraph in page 5: It should be mention that the reversibility of the process has been challenged and that the main substrate of FTO was proposed to be m6Am
4) Last paragraph in page 6: YTHDC1 was also shown to promote nuclear export.
5) Beginning of page 9: m6A also regulates mRNA localization in stress granules.
6) Page 14: the HIV-1 genome is composed of one positive sense RNA that is found as a dimer (two copies of the genomic RNA). As written, it could be interpreted that the HIV-1 genome is segmented in two different RNA molecules
7) Figure 2: Since authors highlight at several points of the manuscript the benefit of the Direct RNA sequencing using Oxford Nanopore Technologies in the determination of m6A at single nucleotide resolution, I suggest to include a scheme of the Direct RNA sequencing strategy in the figure. May be authors should design a new figure showing antibody-dependent, antibody-independent and Direct RNA sequencing determination of m6A.
Author Response
We thank the reviewers for peer-reviewing the manuscript and the kind comments received. We have now addressed their comments in the manuscript and figures to improve the review.
Reviewer 2:
We have now addressed each of the comments from reviewer 2, including an update on figure 2. We have shown the changes with tracked changes on the document.